# A Clinical Guide to Peptide Receptor Radionuclide Therapy with ^177^Lu-DOTATATE in Neuroendocrine Tumor Patients

**DOI:** 10.3390/cancers14235792

**Published:** 2022-11-24

**Authors:** Morticia N. Becx, Noémie S. Minczeles, Tessa Brabander, Wouter W. de Herder, Julie Nonnekens, Johannes Hofland

**Affiliations:** 1Department of Molecular Genetics, Erasmus MC Cancer Institute, Erasmus University Medical Center, 3015 GD Rotterdam, The Netherlands; 2Department of Radiology & Nuclear Medicine, Erasmus MC Cancer Institute, Erasmus University Medical Center, 3015 GD Rotterdam, The Netherlands; 3Department of Internal Medicine, Section of Endocrinology, ENETS Center of Excellence, Erasmus MC Cancer Institute, Erasmus University Medical Center, 3015 GD Rotterdam, The Netherlands

**Keywords:** peptide receptor radionuclide therapy, ^177^Lu-DOTATATE, guide, neuroendocrine tumors

## Abstract

**Simple Summary:**

Peptide receptor radionuclide therapy (PRRT) is one of the treatment options for locally advanced or metastatic gastroenteropancreatic (GEP) neuroendocrine tumors (NETs). This treatment makes use of radioactive labeled somatostatin analogues, with ^177^Lu-DOTATATE as its established standard. PRRT has positive effects in tumor control and it lowers the risk of disease progression or death. It also improves the quality of life of the patient. Unfortunately, important risk factors for a minority of patients include renal and hematological toxicities. NET is a rare disease and treating patients with PRRT requires clinical expertise. This guide gives an overview of the background of PRRT and the current results in NET patient care.

**Abstract:**

Peptide receptor radionuclide therapy (PRRT) with [^177^Lu]Lu-[DOTA^0^,Tyr^3^]-octreotate (^177^Lu-DOTATATE) has become an established second- or third-line treatment option for patients with somatostatin receptor (SSTR)-positive advanced well-differentiated gastroenteropancreatic (GEP) neuroendocrine tumors (NETs). Clinical evidence of the efficacy of PRRT in tumor control has been proven and lower risks of disease progression or death are seen combined with an improved quality of life. When appropriate patient selection is performed, PRRT is accompanied by limited risks for renal and hematological toxicities. Treatment of NET patients with PRRT requires dedicated clinical expertise due to the biological characteristics of PRRT and specific characteristics of NET patients. This review provides an overview for clinicians dealing with NET on the history, molecular characteristics, efficacy, toxicity and relevant clinical specifics of PRRT.

## 1. Introduction

Peptide receptor radionuclide therapy (PRRT) with radiolabeled somatostatin analogs (SSAs) has become an established second- or third-line treatment option for patients with progressive well-differentiated (grade 1–2) gastroenteropancreatic (GEP) neuroendocrine tumors (NETs). Due to the increasing incidence and prevalence of GEP-NETs over recent decades [1] and the development of NET-specific treatments and treatment protocols, there is a growing utilization of such systemic treatment for this advanced patient group. PRRT with [^177^Lu]Lu-[DOTA^0^,Tyr^3^]octreotate (^177^Lu-DOTATATE) is the first registered theranostic application in the field of NETs [2,3]. With this therapy, radiolabeled SSAs target the somatostatin receptor (SSTR) subtype 2 that is over-expressed on the cancer cell surface [4]. Treatment of NET patients with PRRT requires dedicated clinical expertise due to the biological characteristics of PRRT (for example, binding to SSTR, DNA damage induction capacity and the radioactive component) and specific characteristics of NET patients (such as origin, grade and SSTR expression). Consequently, eligibility of NET patients for PRRT should be discussed in an expert multidisciplinary team. This review provides an overview of the development of PRRT, clinical management of patients undergoing PRRT and key unmet needs for future investigations.

## 2. Background of PRRT

### 2.1. Mechanism of Action

PRRT with ^177^Lu-DOTATATE uses an intravenously administered beta radiation-emitting radiopharmaceutical targeting SSTR on tumor cells. SSTRs are G-protein coupled receptors with five subtypes of which subtype 2 (SSTR_2_) is the most commonly expressed in NETs, followed by SSTR_1_ and SSTR_5_ [5,6]. Healthy non-diseased organs can also express SSTR subtypes, as can be seen in the healthy liver, spleen, pituitary, salivary glands, thyroid, adrenals, kidney, prostate, pancreas, intestine and blood vessels [7,8,9]. The antitumoral effect of PRRT is triggered by radiation-induced DNA damage. After binding to the SSTR_2_, the radiopharmaceutical agonists are internalized into the tumor cell where the irradiation induces different types of DNA damage such as DNA single strand breaks (SSBs) and DNA double strand breaks (DSBs) of which the latter are the major contributors to tumor cell death induction [10,11] (Figure 1).

### 2.2. The Choice for ^177^Lu-DOTATATE

Clinical development of PRRT commenced with SSAs labeled with indium-111, yttrium-90 and lutetium-177 [12]. PRRT with ^177^Lu-DOTATATE is approved by the Food and Drug Administration (FDA) and European Medicine Agency (EMA) and consists of the SSA octreotate [13] and 1,4,7,10-tetraazacyclotetradecane-1,4,7,10-tetraacetic acid (DOTA), which is the chelator for stable binding of lutetium-177 to octreotate [14]. Lutetium-177 is a medium-energy beta-emitting radionuclide and a low-energy gamma-emitting radionuclide. The gamma emission allows imaging and dosimetry where the beta emission is used for therapeutic purposes. The beta emission of lutetium-177 shows tissue penetration with a maximum range of 2 mm. Detailed information on chemical structure, production and quality control of ^177^Lu-DOTATATE can be found elsewhere [15]. Indium-111 is much less suitable for therapy compared to lutetium-177, because its auger electrons will not reach the nucleus and will therefore not be able to induce sufficient DNA damage [16]. Another important part of the potential therapeutic effect is the choice of SSA. Octreotate shows a higher affinity to SSTR_2_ than the earlier developed octreotide. This can explain why the uptake of ^177^Lu-DOTATATE in the tumor was increased three to four times in comparison to ^111^In-diethylenetriaminepentaacetic acid (DTPA)-octreotide [17]. Therapeutic activity of ^111^In-DTPA-octreotide, known for its diagnostic use as OctreoScan^®^, originally showed promising preliminary results on symptom control in NETs, but radiological tumor regression was rare and high levels of myelotoxicity were observed [18,19]. Given the specific characteristics of the different radionuclides, not only is the variance in efficacy important to take into account when selecting the preferred radionuclide, but also the risk of toxicity has to be taken into account. Less renal toxicity was seen in patients treated with ^177^Lu-DOTATATE compared to patients treated with [^90^Y]Y-[DOTA^0^,Tyr^3^]octreotide (^90^Y-DOTATOC). This can be explained by the lower beta-energy and tissue penetration of lutetium-177. Due to the lack of gamma emission, yttrium-90 is also not suitable for scintigraphy with a gamma camera [20], however, positron emission tomography (PET) imaging is possible [21]. ^177^Lu-DOTATATE can also be used for tumor dosimetry to obtain more knowledge about the tumor-absorbed dose. An indication of the absorbed dose can be obtained by dosimetric calculation on the total body scintigraphy. Quantitative single photon emission computed tomography/computerized tomography (SPECT/CT) can be more precise in the estimation of the tumor-absorbed dose per cycle. However, these scans are more time consuming, especially for dosimetry at multiple time points. A reduction of uptake in the tumor in every subsequent cycle might be an indication of a good tumor response. On the other hand, there is still a large variety between cycles and different tumor sides within the patient [22]. A significant dose–response correlation was found in pancreatic NETs (panNETs) but not in small intestinal NETs (SI-NETs) [23,24]. The different range of the beta particles from lutetium-177 and yttrium-90 could potentially be an advantage in patients with tumors of different sizes, for instance in an induction setting, although further research is needed [25,26].

## 3. Efficacy of PRRT

### 3.1. Tumor Control

Clinical evidence of the efficacy of PRRT in tumor control was proven by the phase III NETTER-1 study and several phase II clinical trials [27,28,29]. The NETTER-1 clinical trial compared four cycles of 7.4 GBq (800 mCi total) ^177^Lu-DOTATATE plus intramuscular long-acting octreotide 30 mg every 4 weeks to a control group which received octreotide 60 mg every 4 weeks in grade 1 and 2 midgut NET patients, progressive on a long-acting SSA. The rate of progression-free survival (PFS) after 20 months was 65% of patients treated with ^177^Lu-DOTATATE compared to 11% of patients in the control group. The risk of progressive disease (PD) or death was 79% lower in the ^177^Lu-DOTATATE group and the objective response rate (ORR) was significantly higher in the ^177^Lu-DOTATATE group (18%) than in the control group (3%). Sixteen percent of the patients treated with ^177^Lu-DOTATATE showed PD compared to 56% of the control group [27]. After a follow-up of 76 months in the intention-to-treat population, the median overall survival (OS) was 48 months in the ^177^Lu-DOTATATE group, which did not significantly differ from the median OS of 36 months in the control group. The similarity in OS between both groups may be explained by the fact that 36% of the patients in the control group crossed over to the PRRT arm during follow-up [30]. In the largest phase II prospective trial to date, Brabander et al. included patients treated with PRRT with ^177^Lu-DOTATATE for GEP-NETs, bronchial NET and NET of unknown primary origin. A median PFS of 29 months and a median OS of 63 months were observed in 443 patients with bronchial and GEP-NETs who received a cumulative activity of 22.2–29.6 GBq (600–800 mCi) ^177^Lu-DOTATATE. An ORR of 39% after PRRT was found, whereas stable disease (SD) was observed in 43% of the patients [29]. In other phase II trials, retrospective studies and meta-analyses of PRRT with ^177^Lu-DOTATATE, similar survival and response data were found [31,32,33,34,35,36]. An overview of these results is given in Table 1. Although PRRT with ^177^Lu-DOTATATE is the best option for treatment of patients with advanced GEP-NETs at this moment, complete response rates are still rare (1–2%) [27,29].

### 3.2. Symptom Control

PRRT is not only effective in reducing tumor growth, but it can also lead to symptomatic improvement in NET patients, including associated hormonal syndromes [37]. In the NETTER-1 trial, PRRT with ^177^Lu-DOTATATE provided a significant benefit in quality of life (QoL) compared to octreotide LAR. Time to deterioration (TTD) was significantly longer in the ^177^Lu-DOTATATE group, with positive outcomes in the domains of global health, physical functioning, diarrhea, pain, body image, disease-related worries and fatigue [38]. In a prospective series of 265 GEP- and bronchial NET patients, QoL, performance status and symptoms (particularly insomnia, appetite loss and diarrhea) improved significantly after PRRT [39]. Importantly, QoL in asymptomatic NET patients did not decline during therapy. In a single center study of 144 patients, a symptomatic response to PRRT with regard to diarrhea, abdominal pain, flushing and fatigue was observed in 70%, 63%, 64% and 53%, respectively [33]. Results are summarized in Table 1. In patients with a functioning NET, PRRT has shown to be an effective treatment regarding symptom control and circulating hormone levels [40,41]. Carcinoid syndrome (CS) is the most prevalent hormonal NET syndrome and is caused by excretion of hormones and amines such as serotonin, histamine, catecholamines, prostaglandins and tachykinins. In a study involving 22 patients with refractory CS who received PRRT for symptomatic control, flushing and bowel movement frequency improved significantly [42]. Two-thirds of the patients who had at least two episodes of flushing per day had a minimal decrease of 50% of these episodes. Of patients with bowel movement frequency of at least four times a day, 47% experienced > 30% decrease, while 29% experienced > 50% decrease [37]. In a systematic review, symptomatic improvement after PRRT was observed in 74% of patients with diarrhea and in 6% of patients with flushing [43]. Together, these results have positioned PRRT with ^177^Lu-DOTATATE as a viable option for refractory CS [44]. In addition, case reports have shown improvement of symptoms and echocardiographic parameters of carcinoid heart disease after PRRT, which is a severe symptom of CS [45]. Similar positive effects of PRRT on hormonal levels and symptoms have been observed in patients with functioning panNET syndromes, such as insulinoma, gastrinoma, glucagonoma and VIPoma [40].

### 3.3. Prediction of Response to PRRT

There are a variety of NET subtypes with differences in prognosis and response to PRRT. GEP-NETs show higher PFS (25–30 months) and OS (39–71 months) after PRRT than non GEP-NETs (12–29 months PFS and 39–53 months OS) [28,29]. Within the group of GEP-NETs, panNETs and small intestinal NETs are the most prevalent. After PRRT, a median PFS of 30 months for both groups was observed with a median OS of 71 months in panNETs and 60 months in midgut NETs [29]. The outcome of PRRT has also been associated with the extent of tumor burden and in particular the hepatic tumor load [46]. An extensive tumor burden correlated with an increased rate of PD after PRRT and a significantly shorter OS [29,46]. Patients of whom the liver had over 50% tumor burden showed significantly lower OS and PFS in phase II trials [47,48]. Conversely, in the NETTER-1 trial, total liver tumor burden was found not to correlate with PFS, whereas the presence of at least one large target lesion of > 3 cm was associated negatively with PFS [49]. Additionally, higher uptake on SSTR imaging correlated significantly with better response rates after PRRT [50]. Recently, the cumulative activity of ^177^Lu-DOTATATE was found to be an independent predictive factor for survival after PRRT. In 130 patients who received a reduced activity of ^177^Lu-DOTATATE because of disease-unrelated causes (such as bone marrow and renal toxicity), PFS, OS and response rates were all lower as compared to 350 control patients who received 800 mCi ^177^Lu-DOTATATE [51]. The association between PFS and OS and reduced administered activity of ^177^Lu-DOTATATE persisted in multivariable Cox regression analyses in which relevant confounders were included. ^18^F-FDG PET/CT has limited added value in the assessment for PRRT in well-differentiated NETs. Although neuroendocrine neoplasm (NEN) patients with positive uptake on ^18^F-FDG PET/CT had a significantly shorter PFS than patients with a negative scan [52], this modality is likely more a prognostic than predictive marker [53]. A grading score has been developed to integrate the results of both ^18^F-FDG PET/CT and SSA PET imaging ranging from non-aggressive somatostatin receptor imaging (SRI)-positive, FDG-negative to aggressive FDG-positive, SRI-negative NETs. This NETPET score was found to be a significant prognostic marker of OS in patients with GEP-NETs or bronchial NETs, but its predictive role in PRRT is unknown [54,55].

### 3.4. Predictive Markers

The biochemical response to treatment in NETs has often been evaluated by serum chromogranin A (CgA) levels. An increased level is associated with hepatic tumor burden, rapid tumor progression and a shorter OS [56]. In patients who received PRRT, a decrease in CgA of >50% was observed in 25–52% of patients [33,36,57]. Patients with a significant decline in CgA levels after PRRT displayed prolonged PFS and OS [57]. Changes in CgA levels during PRRT should be interpreted with caution though, as an increase in CgA can be caused by both tumor progression and cell damage or lysis by PRRT [58]. The PRRT predictive quotient (PPQ) is a circulating transcript assay that can predict the response of GEP-NETs and bronchopulmonary NETs. A positive PPQ measured before the start of PRRT accurately predicted the outcome of disease control after PRRT in 97% of cases, whereas a negative PPQ was associated with PD after PRRT in 61% of cases at 2 months and 89% at 6–9 months after PRRT [59]. Levels of the NETest, a circulating transcript analysis of 51 genes, during and after PRRT were found to be associated with response according to RECIST on imaging [60]. A lack of decrease in NETest levels at the fourth PRRT cycle signified the advent of PD in advance of imaging, with an overall diagnostic accuracy of 93% to predict response [61].

## 4. PRRT Protocol

The eligibility of a patient to receive PRRT should be discussed in a multidisciplinary team in a specialized NET center for each individual patient. Although multiple protocols for PRRT with ^177^Lu-DOTATATE have been proposed, the most commonly used treatment schedule entails four cycles of 7.4 GBq ^177^Lu-DOTATATE, which is based on the Rotterdam Erasmus MC protocol and also used in the NETTER-1 study [27,29]. The interval between cycles is 6 to 10 weeks. In case of toxicity, this interval can be extended up to 16 weeks [2,3,27]. In parallel with the administration of ^177^Lu-DOTATATE, an amino acid solution of 2.5% arginine and lysine in 1 L saline is co-infused in order to prevent renal toxicity. This infusion starts approximately 30–60 min before the administration of ^177^Lu-DOTATATE with a total infusion time of 4 h. The potential volume overload of the amino acid infusion in decompensated carcinoid heart disease patients can generally be managed by a prolonged infusion period or the administration of loop diuretics [62]. The amino acid infusion can cause nausea for which an antiemetic, typically ondansetron or granisetron, should be given prophylactically before the start of the infusion. ^177^Lu-DOTATATE allows for post-therapy scintigraphy with planar imaging or SPECT/CT. At patient discharge, the radiation exposure should be measured and patients should receive tailored advice on the duration of radiation safety precautions at home, to avoid a high radiation exposure to other people, particularly children and pregnant women. Patients with NET-associated hormonal syndromes who have an indication for continuation of SSA use should adjust the moment of the injections to the PRRT cycles. Long-acting SSA should not be given within 4–6 weeks before a cycle of PRRT because of interference with the radiolabeled SSA. Although there is conflicting evidence from two limited single center studies whether continuation of SSA treatment is beneficial in non-functioning NETs [63,64], this practice is often adopted. If the patient suffers from severe hormonal symptoms, short-acting SSA can be used to bridge this period up till 24 h before PRRT. Radiopharmaceuticals such as ^177^Lu-DOTATATE need to be administered at specialized facilities by medical personnel trained in radiation safety. These facilities should adhere to national and international regulations on the use of radiopharmaceuticals and be licensed by the regulatory authorities. Depending on local protocol and exposure regulations, PRRT with ^177^Lu-DOTATATE can be given in an in-patient as well as an out-patient setting. In between cycles, patients should be reviewed for adverse effects, including full blood count and renal and liver function. Response evaluation by cross-sectional imaging is usually performed 2–3 and 6 months after the last cycle of PRRT. Long-term follow-up is determined on an individual basis taking into account the tumor biology and therapeutic response [65]. Pseudo-progression is a phenomenon that should be considered in the response evaluation when an increase in tumor size is seen during or short after treatment with PRRT. Pseudo-progression is probably based on localized, temporarily edema caused by inflammation as a response to PRRT and does not show the actual tumor response to the therapy [58]. When pseudo-progression is suspected, functional imaging (for example, PET/CT) can help differentiate between true progression and pseudo-progression [31].

## 5. Salvage PRRT

In NET patients who showed tumor response at least 18 months after the first cycle of ^177^Lu-DOTATATE, re-treatment with PRRT (R-PRRT) with two additional cycles of 7.4 GBq each after renewed PD has shown antitumoral effects. In a meta-analysis on the effect of R-PRRT, the pooled median PFS was 14 months with a pooled median OS of 27 months. Similarly, the pooled ORR was 17% with a disease control rate of 77%. Response rates, PFS and OS were lower than for initial PRRT [66], nonetheless R-PRRT remains a potential option for GEP-NET patients when other systemic treatment options are limited. The limited efficacy of R-PRRT as compared to initial PRRT might be explained by the administration of lower cumulative activity (i.e., generally half of the initial PRRT dosage) [51], the increase in tumor bulk at baseline before R-PRRT and potential changes in the tumor biology, such as a longitudinal increase in Ki-67. In the largest study to date by van der Zwan et al., no difference in toxicity after R-PRRT as compared with initial PRRT was observed, particularly no increased occurrence of nephrotoxicity or significant hematological disease [67]. In cases where R-PRRT has provided additional benefit on tumor response and prolonged PFS, further re-treatment at the time of progression can be considered [67].

## 6. Patient Selection

PRRT with ^177^Lu-DOTATATE is registered for patients with GEP-NETs that are progressive on SSA treatment. Non-radiolabeled or ‘cold’ long-acting SSAs such as octreotide LAR and lanreotide are generally considered the first-line systemic treatment option for advanced and metastatic SSTR-positive GEP-NETs, particularly for tumors with a Ki-67 index below 10% [68]. Among the different second-line options in well-differentiated panNETs, ^177^Lu-DOTATATE seems to compare favorably to targeted therapy (everolimus or sunitinib) or chemotherapy [69]. In the randomized phase II OCLURANDUM trial, PRRT with ^177^Lu-DOTATATE was associated with a higher median PFS of 20.7 months compared to 11.0 months for sunitinib treatment in patients with advanced, SSTR-positive, progressive panNET [70]. Currently, the randomized phase III COMPETE trial is comparing PRRT with ^177^Lu-edotreotide with everolimus in advanced, progressive GEP-NET patients. A subset of GEP-NET patients present with extensive tumor bulk or high proliferative rate (Ki-67 index of 10–55%). In these cases, treatment with octreotide LAR or lanreotide has questionable antiproliferative effects [71]. Given its ORR of 39%, which increases to 55% in panNETs [29], ^177^Lu-DOTATATE can be considered as first-line therapy if response is clinically necessitated [72]. These response rates compare favorably to targeted therapy [73,74,75] and for panNETs appear similar to capecitabine–temozolomide chemotherapy [76]. Poorly differentiated neuroendocrine carcinomas (NECs) and well-differentiated grade 3 NETs are high-grade NENs that display a more aggressive biological behavior than the more common grade 1 and 2 NETs [77]. PRRT is currently not considered a standard treatment option for high-grade NENs [68,78]. The rate of SSTR_2_ expression in grade 3 NETs ranges from 67–92% and in NECs from 32–50%, compared to a positive expression rate in grade 1 and 2 NETs ranging from 67–96% [5,79,80,81,82]. In a meta-analysis of PRRT comprising four studies, grade 3 NET patients had a median PFS of 19 months and median OS of 44 months after PRRT. The median PFS was 11 and 4 months and the median OS was 22 and 9 months for NEC with a Ki-67 of 21–55% and NEC with a Ki-67 above 55%, respectively [83]. Recent studies implicated that PRRT could be considered in grade 3 GEP-NETs and GEP-NEC with a Ki-67 of 21–55%. Importantly, to qualify for PRRT, uptake in all lesions is required on somatostatin receptor imaging [83].

## 7. Eligibility Criteria for PRRT

There are several inclusion and exclusion criteria to decide if a patient is eligible for treatment with PRRT. A key criterion for PRRT is the degree of uptake on SSTR imaging which is scored by the Krenning score based on planar ^111^In-DTPA-octereotide imaging. It was reported that ^68^Ga-DOTA-SSA PET/CT results in higher Krenning scores than ^111^In-DTPA-octereotide imaging [84]. The uptake of all tumor lesions should minimally be equal to the physiological uptake in the liver (Krenning score grade 2) on ^111^In-DTPA-octreotide scintigraphy or higher than the physiological uptake in the liver on ^68^Ga-DOTA-SSA PET/CT for a patient to be eligible for PRRT. The latter functional imaging is more accurate for detecting SSTR-positive primary tumors and metastases and therefore superior for assessing the total extent of disease [85,86,87]. PRRT is only applicable for patients with a Karnofsky Performance Scale of at least 60 [27]. PRRT is contra-indicated for patients who are pregnant or breastfeeding, patients with severe cardiac impairment (NYHA III or IV) and those with a life expectancy less than 3 months [31]. Since PRRT can induce toxicity, the following pre-treatment laboratory values are required: creatinine clearance > 40 mL/min, hemoglobin levels ≥ 6 mmol/L, leukocytes > 2 × 10^9^/L, platelet count > 75 × 10^9^/L, bilirubin, alanine aminotransferase (ALAT) and aspartate aminotransferase (ASAT) < 3 times the upper limit of normal and albumin > 3 g/dL [2,3].

## 8. PRRT-Related Toxicity

Adverse events related to PRRT are frequently mild and include nausea, abdominal pain and asthenia [27,50]. Increased hair loss is observed in up to 60% of patients treated with ^177^Lu-DOTATATE, but this is temporary and seldom leads to baldness [88]. Besides these mild adverse events, PRRT can induce more severe toxicities which can be dose-limiting and adjustments to the treatment schedule might be required [89]. When the toxicity has subsided within 16 weeks after the last dose, guidelines advise to administer half of the original activity of ^177^Lu-DOTATATE during the next cycle. PRRT should be discontinued when the toxicity persists after 16 weeks or recurs after the half dose [3]. In the NETTER-1 trial, 7% of the patients received a reduced dose because of dose-limiting toxicities [27]. The kidneys and the bone marrow are the critical organs for dose-limiting toxicities.

### 8.1. Hematological Toxicity

PRRT can induce hematological toxicity through bone marrow radiation. The vast majority of patients only have mild and reversible hematological toxicity with a nadir at 4–6 weeks after administration of PRRT [27,36,50]. However, grade 3 or 4 neutropenia, thrombocytopenia or leukopenia have been observed in, respectively, 1%, 2% and 1% of the patients treated with ^177^Lu-DOTATATE in the NETTER-1 trial [27]. PRRT-induced severe lymphopenia is the most common hematological toxicity [27,29], but it has not been associated with increased susceptibility for infections [90]. Thrombocytopenia is the most common cause of dose reduction in PRRT, whereas bleeding complications are rare [27]. Caution should be taken in patients with widespread bone metastases (Figure 2A), due to the risk of persistent cytopenia. In the absence of alternative treatment options, PRRT should preferably be initiated at half the regular activity (3.7 GBq) ^177^Lu-DOTATATE in these patients. Additionally, there is a relevant long-term risk of 2% for the development of myelodysplastic syndrome (MDS) and 1% for acute myeloid leukemia (AML) after PRRT [27,29,91]. Little is known about the pathophysiology of persistent hematological toxicity, but a role for clonal hematopoiesis has been postulated [92]. Known risk factors for severe hematological toxicity include decreased renal function, pre-existent cytopenias, extensive tumor mass, age over 70, extensive bone metastases and pre-treatment with myelotoxic chemotherapy [91,93,94]. Additionally, women are at higher risk for developing subacute grade ≥ 2 thrombocytopenia than men, which was independent from other risk factors in a multivariable analysis [89].

### 8.2. Nephrotoxicity

Due to SSTR expression in the kidneys and the renal excretion of radiolabeled SSAs, the kidneys receive a high radiation dose during PRRT. Following glomerular filtration, SSAs are re-absorbed in the proximal tubuli of the renal cortex because of active transport mechanisms [95]. By saturating this re-uptake mechanism through the use of lysine and arginine, re-absorption of radiolabeled peptides can be significantly reduced. This results in less radiation-induced nephrotoxicity by a reduction of the absorbed kidney dose up to 40% [96,97]. However, nephrotoxicity after PRRT still occurs such as tubulointerstitial scarring, atrophy and thrombotic microangiopathy [98]. In the NETTER-1 study, grade ≥ 3 renal toxicity was observed in 5% of the ^177^Lu-DOTATATE group and in 4% of the control group [30]. Bergsma et al. reported an overall creatinine clearance loss of 3.4% 1 year after PRRT. No subacute grade ≥ 3 renal toxicity was seen and 1.5% of the patients showed grade 3 renal toxicity in the long term. However, all these patients had a creatinine clearance of <60 mL/min at baseline [99]. A reduced kidney function can lead to a delayed renal excretion of ^177^Lu-DOTATATE and this has also been associated with a higher risk of hematological toxicity [95]. Risk factors associated with renal toxicity include age > 60 years, hypertension, diabetes mellitus, pre-existing renal disease, cumulative radiation dose to the kidneys, previous nephrotoxic chemotherapy, tumor or metastases close to the kidney and previous PRRT with ^90^Y-DOTATOC [100]. Post-renal obstruction can be observed in some GEP-NET patients, particularly in those with retroperitoneal or pelvic metastases, but this can also be caused by the primary tumor, nephrolithiasis and abdominal or retroperitoneal fibrosis [101]. Post-renal obstruction can lead to reduced renal excretion of ^177^Lu-DOTATATE and thereby increase the exposure of radiation to the kidneys, risking (permanent) deterioration of renal function (Figure 2B) [102,103]. In these patients, it is necessary to correct the hydronephrosis of a functional kidney before the start of PRRT to lower the risk of radiation-induced toxicity, for instance with a double J urethral stent or percutaneous nephrostomy [101,104].

### 8.3. Hepatotoxicity

Mild and severe hepatotoxicity has been observed in, respectively, 12% and 0.4–2.5% of patients after PRRT [28,33]. Based on clinical experience, liver failure after PRRT can occur in cases of severe (>90%) liver involvement of NET metastases (example of a patient with severe liver involvement with an enlarged risk of liver failure in Figure 2C). Moreover, there is a concern about the cumulative hepatotoxicity in patients treated with a combination of PRRT and radioembolization. In the HEPAR phase I trial, one patient died of liver failure after treatment with a dose of 80 Gy ^166^Ho-radioembolization [105]. No liver failure was observed in 67 patients undergoing ^90^Y-radioembolization after PRRT [106,107]. Adherence to eligibility criteria is crucial and in borderline cases a first trial infusion with 3.7 GBq ^177^Lu-DOTATATE is advised. There was no relevant hepatotoxicity reported in the NETTER-1 trial [27]. Grade 3 and 4 toxicity of aminotransferases was observed in 3% in the phase II trial by Brabander et al. Three of these patients had persisting toxicity after 3 months, but no hepatic failure was observed [29].

## 9. Hormonal Crisis Due to PRRT

A hormonal crisis can be triggered by PRRT due to excessive release of metabolically active amines or peptides from the tumor. A PRRT-induced hormonal crisis was observed in 1% of patients and can occur during the infusion of the radiolabeled SSA up to 48 h after infusion [108]. Risk factors for a hormonal crisis include CS, elevated 5-hydroxyindoleacetic acid (5-HIAA) and CgA levels, metastatic disease, high tumor burden, higher age and histamine release due to medication as β2 agonist bronchodilators. Prophylactic measures consist of controlling the CS before initiation of PRRT, having a good nutritional status and reducing the time without non-radiolabeled SSAs. When a hormonal crisis occurs, symptom control is necessary. The treatment should be hormone-specific and may also include general measures such as monitoring of vital parameters, saline infusion, correction of electrolyte disturbances and inpatient observation if needed. SSAs can be restarted safely 1 h after infusion of ^177^Lu-DOTATATE and should be considered in severe cases of SSA-responsive CS, gastrinoma, insulinoma and VIPoma. Post-infusion administration of corticosteroids can be considered in patients at risk for carcinoid crisis or those presenting with hormonal crisis [109].

## 10. Nervous System Location or Compression

GEP-NETs rarely metastasize to the central nervous system, with an incidence of brain metastases of 1.5–5% in NET patients [110,111]. More frequently, vertebral bone metastases can compress the spinal cord or its nerve roots (Figure 2D) [112]. PRRT can reduce tumor volume and metabolic activity of central nervous system metastases, thereby reducing pain and spinal compression [113]. The radiation-induced damage by PRRT can be accompanied by temporary edema, leading to increased compression of surrounding vital structures and potentially (worsening of) neurological symptoms. In cases of brain metastases or for those patients at risk of spinal cord or nerve compression, prophylactic therapy with corticosteroids can be considered to prevent or minimize treatment-induced edema. Some expert centers have used a one-week regimen of 4 mg dexamethasone two times a day. Caution is required with this treatment since it is possible that the use of corticosteroids negatively influences the uptake of SSA [114].

## 11. Mesenteric Fibrosis and Mesenteric or Peritoneal Metastases

Another midgut NET-related complication is the presence of mesenteric fibrosis, which occurs in up to half of the patients with a small intestinal NET metastatic to the mesenteric lymph nodes. Mesenteric metastases can induce local fibrosis by the secretion of serotonin and other mediators, inducing a desmoplastic reaction, which subsequently can lead to intestinal obstruction and ischemia (Figure 2E) [115,116]. Peritoneal metastases can lead to adhesions between the abdominal wall and the bowel which can cause intestinal immobility, bowel obstruction or ‘frozen abdomen’ (Figure 2F) [117]. The proportions of patients treated with PRRT who had mesenteric and/or peritoneal metastases ranged from 6–51% [27,117,118]. In a retrospective study of 132 NET patients with a mesenteric mass, ^177^Lu-DOTATATE resulted in an ORR of this mass in only 4% of the patients [119]. In an analysis of NET patients with mesenteric or peritoneal disease, 23% of patients at high risk for complications experienced at least one episode of bowel obstruction within three months of PRRT. These patients were treated with corticosteroids, surgery and total parenteral nutrition [117]. Although a direct causal relationship has not been established, treating physicians should be aware that PRRT seems to be ineffective in reducing the size of a mesenteric mass while there may be an increased risk of abdominal complications in patients with extensive peritoneal disease or desmoplastic changes associated with mesenteric metastases. A post-infusion trial of corticosteroids can be considered in patients at high risk of these complications [117].

## 12. Conclusions and Future Directions

PRRT with ^177^Lu-DOTATATE is a widely applicable therapy for patients with advanced and metastatic NETs. It is clear that PRRT contributes to reduction of tumor growth and stabilization of the disease in a significant proportion of these patients. Next, PRRT improves QOL and reduces symptoms [27,29]. Despite the successes, it is important that clinicians treating NET patients are aware of the toxicities (mainly renal and hematological) associated with PRRT and discuss every patient in a multidisciplinary team where careful selection, preparation and management of these patients before and during therapy with this unique radiopharmaceutical can be discussed. Despites its successes, there still is room for improvement of the efficacy and safety of PRRT. Ongoing research projects are exploring how to improve the current strategies. A combination therapy with radiosensitizers is a way to increase the effect of PRRT. This can be conducted by the inhibition of poly ADP-ribose polymerase (PARP)-1 which is essential for the repair of SSBs [120] or inhibition of heat shock protein 90 (HSP90), a chaperone molecule that plays a role in cell protection and maintenance [121]. Other radionuclides are being studied for potential use in PRRT, such as actinium-225, lead-212 and bismuth-213. These alpha-emitting radionuclides can potentially cause more damage to the tumor cells compared to PRRT with ^177^Lu-DOTATATE and thus increase treatment efficacy [122,123,124]. Altogether, PRRT with ^177^Lu-DOTATATE is a very potent treatment modality for patients with progressive well-differentiated (grade 1–2) GEP-NET and there are many different strategies currently being explored that can contribute even to better patient care.

## Figures and Tables

**Figure 1 cancers-14-05792-f001:**
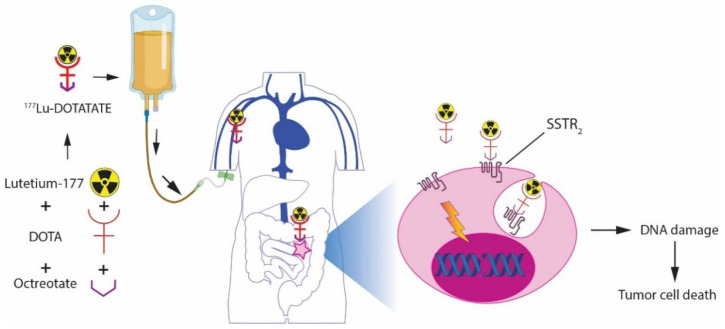
Mechanism of action of PRRT with ^177^Lu-DOTATATE. Intravenous administration of ^177^Lu-DOTATATE leads to tumor cell binding via SSTR_2_. After internalization of the radiopharmaceutical–SSTR_2_ complex, local radiation by beta particles can lead to cell death through the induction of DNA damage (image created with BioRender.com, accessed on 1 September 2022).

**Figure 2 cancers-14-05792-f002:**
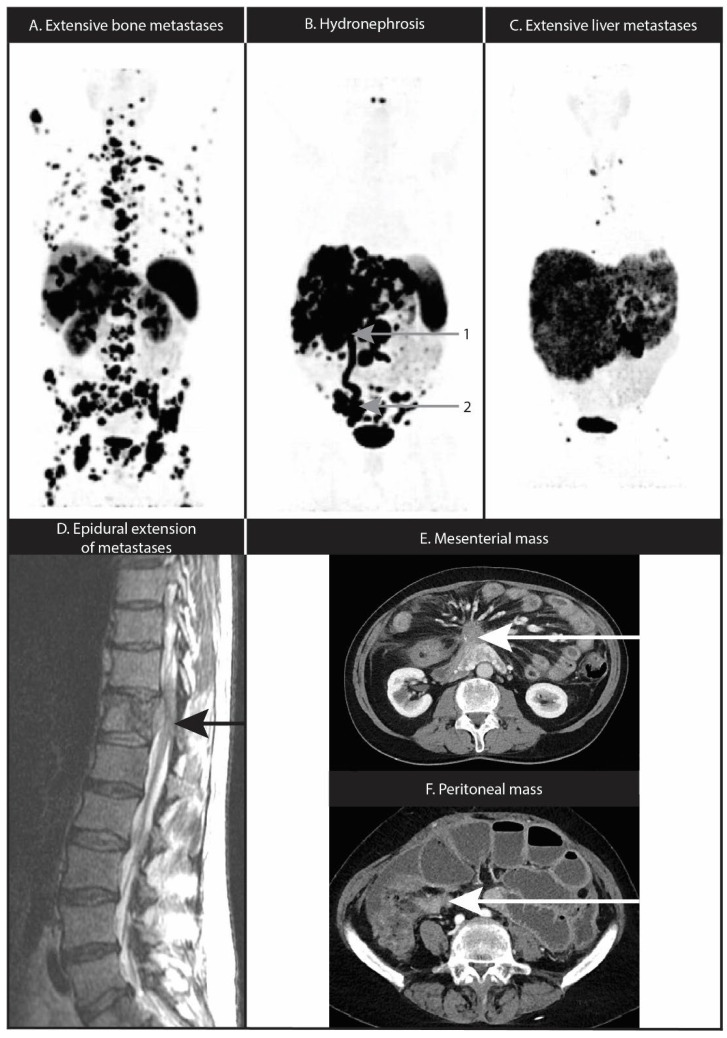
Potential complicated clinical situations for PRRT. (**A**) ^68^Ga-DOTATATE PET scan of extensive bone metastases of SI-NET. R-PRRT with reduced activity of 3.7 GBq ^177^Lu-DOTATATE induced hematoxicity. (**B**) Hydronephrosis (1) due to obstruction by peritoneal tumor deposit of metastatic panNET (2), leading to accumulation of radioactivity in renal medulla. (**C**) ^68^Ga-DOTATATE PET scan showing extensive liver metastases of panNET before treatment with 3.7 GBq ^177^Lu-DOTATATE, which was well tolerated by the patient. (**D**) Bone metastasis of panNET with epidural extension at Th12 on MRI. Post-PRRT edema can lead to infringement of the spinal cord. (**E**) Patient with small bowel NET and metastatic mesenteric mass with desmoplastic reaction, resulting in venous congestion, bowel wall thickening and ascites. The patient suffered from intermittent abdominal pain with temporary aggravation of complaints during PRRT. (**F**) Contrast enhanced CT scan of peritoneal metastases of a multifocal NET of a patient who developed a paralytic ileus or ‘frozen abdomen’ after PRRT.

**Table 1 cancers-14-05792-t001:** Overview of tumor control and symptom control following PRRT as described in key phase II and III clinical trials.

	Patients, *n*	NET Subtype	PFS (Months)	mOS (Months)	ORR	SD	PD	mTTD QOL Global Health (Months)	Overall Symptom Improvement
**Strosberg et al. [27,30,38]**	101	SI-NET	28	48	18%	66% *	16%	29 m	
**Control group SSA**	100	SI-NET	8.5	36	3%	41%	56%	6.1 m	
**Brabander et al. [29]**	443	GEP and bronchial NET	29	63	39%	43%	12%		
**Bodei et al. [36]**	51	GEP and bronchial NET	36	68% at 36 months	55%	27%	18%		
**Ezziddin et al. [34]**	68	GEP-NET	34	53	72%	13%	15%		
**Sabet et al. [35]**	61	SI-NET			44%	48%	8%		
**Hamiditabar et al. [33]**	143	NET of all origins			8%	46%	38%		47%
**Khan et al. [39]**	265	GEP and bronchial NET							38–90%

PFS = progression-free survival, mOS = median overall survival, ORR = objective response rate, SD = stable disease, PD = progressive disease, mTTD QOL = median time to deterioration in quality of life, m = months. * Estimated percentage based on calculations.

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
