# Peer review of "A Clinical Guide to Peptide Receptor Radionuclide Therapy with 177Lu-DOTATATE in Neuroendocrine Tumor Patients"

_cancers, 2022, doi:10.3390/cancers14235792_

Round 1
Reviewer 1 Report
An overall a very well written review on this important topic, well designed and comprehensive- a very good addition to the literaure. .
A few minor comments for consideration by the authors:
1. Introduction: Add comment re the decision re PRRT must be based on a dedicated NET PRRT MDT
2. Line 92: There has been small studies of the use of 90-Ytrium in combination with Lutate as induction therapy in patients with large volume disease,
3. EFFICACY LINE 119: A table summarising the efficacy and symptom benefits of the phase III and selected phase II trials would be of benefit to the reader.
4. Line 129: include the Stable Disease rates for the NETTER-1 trial
4. Line 195- comment on utility of FDG PET to exclude patients with discordant disease esp those whose tumours have a Ki67 >10% - hence not suitable for PRRT
5. Line 208: details of the PPQ – what is the transcript? Have the PPQ and NETest been validated? Comment on the NETPET score.
6. Line 217 in patients with secretory syndromes comment re the risk of hormonal flare post PRRT and hence the need for inpatient observation.
7. Line 306- is there a preferred Krening score for PRRT selection
Author Response
"Please see the attachment."

Reviewer 2 Report
This article reviews 177Lu-DOTATATE-based PRRT and its clinical applications. The benefits including tumor control, symptom control, prediction of response, predictive markers. The drawbacks include PRRT-related toxicity (hematological, nephrological and hepato-toxicity). PRRT protocol, patient selection, and eligibility criteria are also described. The results can provide guidance for further clinical applications and improvements. Comments and suggestions: 1) It is interesting to include a paragraph for describing the chemical structure, production, and quality control of 177Lu-DOTATATE for clinical applications; 2) It is interesting to introduce some results on molecular imaging-based targeting, localization, and distribution of 177Lu-DOTATATE for better understanding of its effects and toxicity. Just like 68Ga-DOTATATE for PET imaging, 177Lu-DOTATATE can be used for molecular imaging by SPECT as reported by others.
Author Response
"Please see the attachment."
